# Determinants of Generative AI System Adoption and Usage Behavior in Korean Companies: Applying the UTAUT Model

**DOI:** 10.3390/bs14111035

**Published:** 2024-11-04

**Authors:** Youngsoo Kim, Victor Blazquez, Taeyeon Oh

**Affiliations:** 1Seoul Business School, aSSIST University, Seoul 03767, Republic of Korea; yssskim1@stud.assist.ac.kr; 2Department of Business Economics, Health and Social Care, University of Applied Sciences and Arts of Southern Switzerland, 6928 Manno, Switzerland; victor.blazquez@supsi.ch; 3Taylor Institute, Franklin University Switzerland, 6924 Sorengo, Switzerland; 4Seoul AI School, aSSIST University, Seoul 03767, Republic of Korea

**Keywords:** generative AI, technology adoption, adoption factors, UTAUT model, South Korean companies

## Abstract

This study addresses the academic gap in the adoption of generative AI systems by investigating the factors influencing technology acceptance and usage behavior in Korean firms. Although recent advancements in AI are accelerating digital transformation and innovation, empirical research on the adoption of these systems remains scarce. To fill this gap, this study applies the Unified Theory of Acceptance and Use of Technology (UTAUT) model, surveying 300 employees from both large and small enterprises in South Korea. The findings reveal that effort expectancy and social influence significantly influence employees’ behavioral intention to use generative AI systems. Specifically, effort expectancy plays a critical role in the early stages of adoption, while social influence, including support from supervisors and peers, strongly drives the adoption process. In contrast, performance expectancy and facilitating conditions show no significant impact. The study also highlights the differential effects of age and work experience on behavioral intention and usage behavior. For older employees, social support is a key factor in technology acceptance, whereas employees with more experience exhibit a more positive attitude toward adopting new technologies. Conversely, facilitating conditions are more critical for younger employees. This study contributes to the understanding of the interaction between various factors in AI technology adoption and offers strategic insights for the successful implementation of AI systems in Korean companies.

## 1. Introduction

With the advent of the Fourth Industrial Revolution, artificial intelligence (AI) has come to play a vital role in increasing productivity and improving operational efficiency in various industries, including manufacturing, services, and finance, and is rapidly becoming a critical technology that must be adopted and promoted in the modern business environment [1,2]. In particular, generative AI and the generative AI systems that enterprises utilize are technologies that automatically generate content in various forms, such as text, images, music, and video, and have great potential to maximize work efficiency and creativity. For example, models such as OpenAI’s ChatGPT are being utilized in various fields such as customer service, marketing, and data analysis due to their ability to generate natural language and images [3,4]. Specifically, generative AI has been used in software to automatically generate program code and detect bugs, reducing development time and costs; in visualization as a tool to effectively analyze complex data sets and automatically generate visuals; and in education to create a learner-centric educational environment using personalized learning materials and simulations to improve student engagement and learning outcomes significantly [5,6,7]. Generative AI technologies continue to drive innovation in various industries, and their applications and potential are expected to expand. Furthermore, artificial intelligence plays a critical role in corporate innovation by facilitating new product development, improving operational efficiency, and providing innovative insights through data analysis [8,9,10].

Generative AI also has the potential to validate subjective findings in research, helping to understand consumer experiences and providing insights beyond human intuition [11]. Despite the rapid advancement and growing adoption of AI technologies, a significant gap persists between users’ intentions to adopt such technologies and their actual usage behavior. This disconnect remains a prominent and unresolved theoretical issue in academic discourse. While recent studies have examined the factors that influence AI adoption intentions, there is still a notable lack of theoretical integration regarding the adoption of emerging technologies like generative AI. This study aims to address this gap by empirically investigating the factors that significantly influence both the intention and behavior of generative AI system adoption. Specifically, it seeks to identify how key determinants, including performance expectancy, effort expectancy, and social influence shape employees’ acceptance and continued usage of these systems in Korean companies. The necessity of this research lies in its potential to provide organizations with actionable insights into optimizing AI adoption strategies, particularly as companies navigate the complexities of integrating generative AI systems into their operations. By applying the UTAUT model, this study contributes to filling a gap in the current literature by exploring the interaction between adoption intentions and post-adoption behaviors.

However, the adoption of generative AI systems is a challenging process. While many companies are making the decision to adopt generative AI systems, it is often done inefficiently and piecemeal, as technical, organizational, and environmental factors work together to make the decision more uncertain and complex [4,9]. In addition, lack of technical infrastructure, difficulty in user acceptance, lack of resources within the organization, compatibility issues with existing systems and processes, and prolonged decision-making are significant obstacles for companies to adopt and use the system [12,13]. In particular, small and medium-sized enterprises face many difficulties in effectively adopting and utilizing generative AI systems due to a lack of resources and expertise compared to large enterprises [14]. Therefore, when adopting generative AI systems, technical, organizational, and social factors should be considered, as well as legal issues such as copyright issues and social acceptance [15].

Furthermore, there are other challenges to adopting generative AI technologies, such as ethical issues and uncertainty [16]. Research is needed to measure and address uncertainty surrounding the technology to manage the quality and unintended consequences of the content generated by generative AI systems. In addition, relevant training, regulations, and governance measures should be put in place to ensure the transparent and responsible use of generative AI systems.

This study leverages the well-established Unified Theory of Acceptance and Use of Technology (UTAUT) model to systematically examine the adoption, usage intentions, and behavioral patterns surrounding generative AI systems, considering their various advantages and challenges. Despite the growing body of research on AI system adoption, there remains a lack of studies examining the interaction between technology acceptance intentions and actual usage behavior post-adoption. Particularly in the case of cutting-edge technologies like generative AI system, research on the complex factors influencing adoption is still insufficient. This study aims to address this gap by empirically analyzing how critical variables function during the AI system adoption process, thus contributing to the theoretical understanding of recent AI technologies. The UTAUT model offers a robust theoretical framework for analyzing key determinants of technology acceptance, encompassing four core constructs: performance expectancy, effort expectancy, social influence, and facilitating conditions [17]. This framework remains highly relevant even during the initial stages of technology adoption, where organizational members develop expectations regarding performance and effort, and make adoption decisions influenced by social and organizational factors. Particularly in the early adoption phase of generative AI systems, where clear expectations of technology performance are yet to be fully established, it becomes critical for users to gain experience through continuous learning and adaptation. While UTAUT2 includes consumer-centric variables such as price value and enjoyment expectations, the focus of this study is on employee adoption of technology within the enterprise, so the UTAUT model was deemed more appropriate.

This study aims to better understand the behavioral intention and usage behavior of generative AI systems by applying the UTAUT model to a survey of 300 employees in Korean companies. By addressing the central research question, “What factors most significantly influence the adoption and use behavior of generative AI systems in Korean companies?”, this study aims to provide actionable insights that help companies develop strategies to support successful adoption and utilization from the outset. By thoroughly understanding the critical determinants of user acceptance, companies are well-positioned to strengthen their competitive advantage and optimize the overall impact of technology integration.

## 2. Research Methodology

The Unified Theory of Acceptance and Use of Technology (UTAUT) model was proposed by [17] and has been widely used to predict technology acceptance and use behavior. The model is widely cited in research in many fields, including information systems, healthcare, and education, and has been utilized in thousands of papers. For example, ref. [18] highlighted that the UTAUT model has been extensively validated and used in empirical research. The UTAUT model includes four main components: performance expectancy, effort expectancy, social influence, and facilitating conditions, which are essential influencers of technology acceptance.

Performance expectancy refers to a user’s belief that their job performance will be improved by using a particular technology, which has been proven in various studies to be one of the most influential factors of technology adoption intention [17,19]. For example, ref. [20] found that performance expectancy is a strong predictor of users’ intention to adopt a technology, and more recent studies have also identified performance expectancy as a vital factor [18]. Accordingly, we hypothesize (H1) that performance expectancy will have a positive effect on behavioral intention.

Effort expectancy refers to a user’s perception of the amount of effort required to use a particular technology, and the easier the technology is perceived to be to use, the higher the intention to use it [19]. A study by [17] found that effort expectancy plays a vital role in technology acceptance, and recent research supports this [21,22]. Therefore, we hypothesize (H2) that effort expectancy will positively affect behavioral intention.

In addition, social influence refers to social pressures or expectations for technology use, including the influence of supervisors, peers, organizational culture, etc., on technology acceptance [23]. Recent studies have also identified it as an important variable [24]. Accordingly, we hypothesize (H3) that social influence will have a positive effect on behavioral intention.

Facilitating conditions refer to the existence of organizational and technical infrastructure that supports the use of a technology and are considered essential for successful technology adoption [25,26]. Ref. [27] also found facilitating conditions to be a significant predictor; therefore, we can propose hypothesis (H4), that facilitating conditions will positively affect behavioral intention.

Moreover, generative AI is a technology that generates new content based on trained data and includes tools that generate output based on user-input commands, such as text, code, or images [10,28]. These technologies are revolutionizing many fields, especially in the software industry, where they have the potential to improve productivity significantly [5]. For example, tools such as Gemini 1.5, GPT-4o, and Microsoft 365 Copilot automate code generation, bug fixing, and documentation tasks, allowing developers and users to focus on more creative and complex problem-solving [5]. Generative AI is used not only to generate text using large-scale language models but also to generate other forms of content, such as images, music, and prototypes [29].

In addition, generative AI systems are comprehensive software and hardware solutions that utilize these generative AI technologies to achieve specific goals [4]. Organizations can integrate these systems with robotic process automation (RPA) and traditional AI technologies to reduce repetitive tasks and focus on high-value work [30]. For example, generative AI like ChatGPT can help executives make decisions more efficiently and improve business strategy through AI-powered analytics. While large enterprises have sufficient resources and technically skilled personnel to adopt such systems quickly, SMEs need more resources and expertise [14]. However, SMEs can also expect to improve their business actively if they adopt these systems. Therefore, we hypothesize (H5) that behavioral intention will positively affect usage behavior.

When considering the mediating effect of behavioral intention on the effects of performance expectancy, effort expectancy, social influence, and facilitating conditions on usage behavior, various studies have demonstrated that behavioral intention is an important mediator. Ref. [17] found that performance expectancy, effort expectancy, social influence, and facilitating conditions combine to influence behavioral intention, which influences usage behavior. Ref. [31] found that these variables significantly impact technology use intention and behavior. Ref. [18] suggested that social influence and facilitating conditions are essential in determining intention to use and usage behavior. Therefore, we propose the hypothesis (H6) that there will be a significant mediating effect of performance expectancy, effort expectancy, social influence, and facilitating conditions on the effect of behavioral intention on usage behavior.

Age can play a role in technology adoption and use. For example, older employees may have more negative attitudes toward technology adoption. Refs. [32,33] found that older employees may resist accepting new technology, emphasizing that age is an essential variable in technology adoption. This suggests the need for technology training programs targeted at older employees and that age differences should be considered in technology adoption strategies. In addition, the ref. [34] study using the UTAUT model concludes that understanding the differences in technology acceptance by age highlights the need for a customized approach and suggests that users of different ages need tailored support and training. Therefore, we hypothesize that age will have a moderating effect (H7).

Work experience can also play a significant role in technological adoption and use. Many studies have shown that employees with more work experience are more willing to use technology. Ref. [35] found that employees with more work experience responded more positively to adopting AI-based technologies because they better understand the technology’s usefulness and ease of use. Refs. [32,36] also found that employees with more work experience have less resistance and more positive attitudes toward technology. Furthermore, ref. [37] demonstrated that AI adoption can either support or hinder proactive career behavior depending on the level of organizational support, with more experienced employees benefiting more from such support. Therefore, we hypothesize that work experience will have a moderating effect (H8).

Based on the above hypothesis, this study aims to analyze the influence on the behavioral intention to use generative AI systems and the usage behavior of Korean companies. The research model is schematized as shown in Figure 1 below.

This study aims to apply the UTAUT model to systematically analyze the effects of performance expectancy, effort expectancy, social influence, and facilitating conditions on the behavioral intention to use generative AI systems and actual usage behavior in Korean companies. Age and work experience are set as moderators, behavioral intention as a mediator, and technology usage behavior as the dependent variable. A quantitative research method was employed, using a survey to collect and analyze the data.

### 2.1. Participants and Sampling

The survey for this study was conducted among 300 employees of Korean companies in various industries. The sample included a wide range of positions and roles in large and small companies to increase the generalizability of the findings and reflect different perspectives on adopting generative AI systems [38]. The sample was selected using a random sampling method, including respondents from various backgrounds and job roles. The survey was conducted online, and respondents’ personal information was anonymized [39].

### 2.2. Survey Design and Variables

The survey for this study was conducted in June 2024 among employees working in large and medium-sized enterprises in Korea. A total of 354 data points were collected from the online survey, of which the final 300 were used for analysis after excluding missing values and non-responses. The demographic characteristics of the participants in this study are shown in Table 1 below.

The variables of performance expectancy, effort expectancy, social influence, facilitating conditions, behavioral intention, and usage behavior selected in this study are based on several previous studies. The operational definitions of the variables used are shown in Table 2 and refer to the work of [40].

### 2.3. Data Collection and Bias Control

The questionnaire (please see the Appendix A) was organized to reflect the main components of the UTAUT model. Each construct measures performance expectancy, effort expectancy, social influence, facilitating conditions, behavioral intention, and usage behavior. Each item was designed to be answered on a 7-point Likert scale [17]. The study was based on survey response data and took several measures to minimize the problem of common method bias (CMB): first, the survey questions were randomized to prevent respondents from unconsciously answering in a consistent manner; second, the online survey was conducted by a professional research firm and only one question was asked per response, encouraging respondents to rate each question independently. This design helped reduce the likelihood that respondents would answer all questions the same.

Additionally, we examined the impact of Common Method Bias (CMB) using [44]’s single-factor test. Based on [44], the analysis showed that the variance explained by a single factor did not exceed 50%, indicating that there was no concentration on a single factor. These measures demonstrate that our findings are free from CMB.

### 2.4. Statistical Analysis Methods

To accomplish the analytical objectives of this study, SPSS 22 and AMOS 21 were employed. Descriptive statistics and bivariate correlation analyses were first conducted in SPSS 22 to explore the basic interrelationships among variables. Reliability analysis was conducted, adopting a Cronbach’s Alpha value of 0.6 or higher, as suggested by [45]. Although exploratory factor analysis (EFA) was initially used to preliminarily align factor structures with the theoretical framework, the primary analytical focus was on confirmatory factor analysis (CFA) carried out in AMOS 21. Upon confirming multivariate normality, CFA was executed via the Maximum Likelihood method to assess construct validity, with path coefficients examined to evaluate model fit. For model fit assessment, the hypothesized model, which incorporates performance expectancy, effort expectancy, social influence, and facilitating conditions, was compared against a null model with no assumed relationships among latent variables, ensuring that the hypothesized model better represented the data and corresponded to the study’s theoretical constructs.

To verify the model’s validity further, convergent and discriminant validity were evaluated through Average Variance Extracted (AVE) and Composite Construct Reliability (CCR). The indirect effects of usage intention were analyzed through bootstrapping (500 resamples, with *p* < 0.05), and the moderating effects of age and work experience were assessed using Multi-Group Analysis (MGA). All statistical analyses followed a significance level of *p* < 0.05.

Additionally, detailed fit indices were reported for both the null model and the hypothesized model. Fit indices such as Chi-square (χ^2^), RMSEA, CFI, TLI, and RMR were analyzed to clearly demonstrate the improvements in the hypothesized model. These results underscore the alignment of the hypothesized model with theoretical expectations, validating that the model integrating performance expectancy, effort expectancy, social influence, and facilitating conditions offers a more accurate and meaningful fit than the null model.

In this study, structural equation modeling (SEM) was used to test the hypotheses and analyze the relationships between the main factors. According to [46], a sample size of 200 or more is generally recommended for structural equation models, and 300 or more samples are sometimes required for complex models. Ref. [47] noted that using 300 or more samples in SEM analysis can contribute to increasing statistical power and model fit.

Following these recommendations, this study used a sample of 300, which meets the appropriate sample size for SEM analysis. In addition, ref. [48] recommends that a sample size of at least 10 subjects per factor is required for SEM analysis, and the sample size of this study fully meets this criterion. According to [49]’s guidelines, a sample size of 300 is sufficient to achieve a statistical power of 0.8 or higher. Therefore, this study used an appropriate sample size for SEM analysis.

## 3. Results

The descriptive statistical analysis results of the measurement variables used in this study are shown in Table 3 below. The mean of performance expectancy was 5.34 ± 0.89, effort expectancy was 4.84 ± 0.95, social influence was 4.30 ± 1.18, facilitating conditions were 3.81 ± 1.39, behavioral intention was 4.90 ± 1.17, and usage behavior was 4.45 ± 1.43, confirming that the mean of performance expectancy factor was the highest. On the contrary, facilitating conditions were the lowest. Structural equation modeling (SEM) was used in data analysis to clarify the interaction between variables and analyze the complex relationship structure [46]. The absolute values of skewness and kurtosis for the factors used in this study were analyzed and found to be below the thresholds of 2 and 7, respectively, thus fulfilling the assumption of normality [50].

In addition, we collected and analyzed data on respondents’ experiences with the adoption and use of generative AI systems. Among the respondents, 20.7% reported having used an AI system in their workplace, 70.0% had not, and 9.3% were uncertain. Regarding the specific AI systems utilized, 48.0% had experience with the free version of ChatGPT, 5.0% with the paid version, and 1.7% with Gemini.

Furthermore, an analysis of the impact of AI system usage on work performance revealed that respondents who had utilized AI systems reported an average score of 5.60 for performance efficiency, 4.92 for performance improvement, 5.31 for competitive advantage, and 5.40 for cost reduction. These findings suggest that the use of AI systems can positively influence an organization’s competitive positioning and cost efficiency. Overall, the results underscore the potential of generative AI systems as a critical technological asset, not only for enhancing operational performance but also for supporting the achievement of broader strategic objectives.

This study comprises four independent variables, one mediator, and one dependent variable. To rigorously assess the reliability and validity of each construct, multiple analyses were conducted. Initially, an exploratory factor analysis (EFA) was performed on the independent variables (performance expectancy, effort expectancy, social influence, and facilitating conditions). Given the large number of items within the independent variables, the analysis was focused solely on these variables to account for potential item exclusion.

In addition, as a result of the exploratory factor analysis of the independent variables selected in this study, for performance expectancy, effort expectancy, social influence, and facilitating conditions, the Eigenvalue ranged from 3.803 to 7.227, which is above the criterion of 1.0, as shown in Table 4; all variables were classified, and the cumulative variance was found to be 70. In addition, the factor loading was more significant than the criterion of 0.4 for all items, which verified the convergent validity between the measured variables of the same factor; the KMO was 0.939, and the chi-square value was 8503.318 (df = 465, *p* < 0.001) in Bartlett’s sphericity test.

In detail, social influence was analyzed with ten items (explained variance = 23.312%, Eigen = 7.227), and facilitating conditions were analyzed with seven relevant items (explained variance = 17.796%, Eigen = 5.517). The scale consisted of seven related items, each for performance expectancy (explained variance = 17.286%, Eigen = 5.359) and effort expectancy (explained variance = 12.266%, Eigen = 3.803). The reliability analysis yielded Cronbach’s Alpha values ranging from 0.891 to 0.956, exceeding the threshold of 0.6.

For the confirmatory factor analysis (CFA), all four independent variables of the research model, along with the mediator (behavioral intention) and the dependent variable (usage behavior), were included. Based on this, correlation analysis, average variance extracted (AVE), and composite construct reliability (CCR) were conducted. Additionally, two exploratory factor analysis results were identified, as shown in Table 5. The Eigenvalues of 1.714 and 1.747 were above the criterion of 1.0, indicating that the factors were accurately classified. The cumulative variance was found to be 86.509%, and the factor loading was more significant than the criterion of 0.4, which verified both the convergent validity of the measures of the same factor. In detail, behavioral intention and usage behavior were separated into two related items. The KMO was 0.767, Bartlett’s test of sphericity revealed a chi-square value of 666.985 (df = 465, *p* < 0.001), and the reliability analysis yielded a Cronbach Alpha of 0.811 and 0.858.

The convergent validity of the factors selected in this study was demonstrated through exploratory factor analysis and reliability analysis. However, confirmatory factor analysis and discriminant validity analysis were conducted to confirm the single dimension of each factor for the measurement items, and the results are shown in Table 6. Compared to the indicators of Model Fit, *χ*^2^ (df = 416, *n* = 300) = 1021.80, *p* = 0.000, CMIN/df = 2.456. CFI = 0.929, RMR = 0.099, TLI (Tucker-Lweis) = 0.921, IFI = 0.929, and RMSEA = 0.070, confirming that the structural model fit for the research model set in this study meets the criteria. In addition, the average variance extracted (AVE) of this study was selected to be above 5 [51] and the composite construct reliability (CCR) was selected to be above 7 [52,53]. The results of the analysis showed that they all met the criteria, confirming the discriminant validity of the measurement tools.

In addition, as a result of confirmatory factor analysis, the SMC values of performance expectancy No. 3, effort expectancy No. 2, 3, and 7 were found to be below the criterion value and were deleted. The size of the critical ratio (C.R.) of the structural model estimation after deletion was interpreted as an absolute value of 1.96 or more, as shown in Table 6. As shown in Table 6, the C.R. values of the measured variables were found to be above the criterion of 1.96 and significant at *p* < 0.001, so it is judged that the convergent validity of this research model is proven.

As shown in Table 7, the results of the correlation analysis of the factors selected in this study show that performance expectancy has a significant positive correlation of r = 0.614 (*p* < 0.001) with effort expectancy, r = 0.466 (*p* < 0.001) with behavioral intention, r = 0.392 (*p* < 0. 001) with behavioral intention, r = 0.500 (*p* < 0.001) with social affect, r = 0.419 (*p* < 0.001) with social influence, and r = 0.375 (*p* < 0.001) with behavioral intention. In addition, social influence was positively correlated with facilitating conditions at r = 0.725 (*p* < 0.001), facilitating conditions were positively correlated with usage behavior at r = 0.608 (*p* < 0.001), and behavioral intention at r = 0.704 (*p* < 0.001).

Based on the above statistical results, we will test the hypotheses related to the main variables of performance expectancy, effort expectancy, social influence, and facilitating conditions. The analysis results using path coefficients to identify the hypotheses set in this study are shown in Table 8. First, for the structural model fit, *χ*^2^ (df =420, n = 300) = 1066.84, *p* = 0.000, CMIN/df = 2.540. CFI = 0.924, RMR = 0.112, TLI (Tucker-Lweis) = 0.916, IFI = 0.925, and RMSEA = 0.072, which is in line with the structural model criteria.

The results of the detailed analysis showed that there was no significant effect of performance expectancy on behavioral intention with *β* = 0.072 (*p* = 0.297), or of effort expectancy on behavioral intention with *β* = 0.174 (*p* < 0.05), which was significant and positive. Hypothesis 3, the effect of social influence on behavioral intention, was analyzed as *β* = 0.662 (*p* < 0.001), which was found to have a positive effect. For Hypothesis 4, the effect of facilitating conditions on behavioral intention, there was no significant effect (*β* = 0.037, *p* = 0.606).

We tested Hypothesis 5, which states that the parameter behavioral intention affects usage behavior, and the result is *β* = 0.863 (*p* < 0.001), confirming a positive effect. Taken together, these results suggest that as effort expectancy and social influence increase, behavioral intention increases, and as behavioral intention increases, usage behavior increases. In addition, the most influential interaction was identified as the interaction between behavioral intention and behavior, followed by the interaction between social influence and behavioral intention. Based on these results, research hypotheses H2, H3, and H5 are accepted, and H1 and H4 are rejected.

The results of the analysis using bootstrapping to verify the mediating effect of behavioral intention, a parameter of this study, are shown in Table 9. The results indicate that social influence has a significant indirect effect on usage behavior through behavioral intention, with a 95% confidence interval between 0.355 and 0.806 (*p* < 0.01). This finding demonstrates that social influence significantly contributes to usage behavior when mediated by behavioral intention, supporting Hypothesis 6 as partially accepted. However, no significant indirect effects were observed for performance expectancy, effort expectancy, and facilitating conditions.

To further clarify the effect pathways, Hypothesis 5, which posits that behavioral intention has a direct effect on usage behavior, was confirmed with a strong and statistically significant outcome (*β* = 0.863, *p* < 0.001). This suggests that behavioral intention has a substantial direct impact on usage behavior, independent of any mediating variables. Thus, as employees’ behavioral intention to use generative AI system strengthens, their actual usage behavior proportionally increases.

These findings highlight the importance of both direct and indirect pathways, where the direct effect of behavioral intention on usage behavior plays a primary role, while the indirect pathway through social influence emphasizes the critical role of external social factors in the early stages of generative AI system adoption.

Finally, we examined the moderating effects of age and work experience. The moderating effects depicted in Figure 1 reflect the differences between these two variables. Accordingly, H7 tests the moderating effect of age, with the results presented in Table 10, while H8 focuses on the moderating effect of work experience, as shown in Table 11. These two hypotheses were not combined into a single table due to the limitations of Multi-Group Analysis (MGA), which requires each variable to be analyzed independently. Therefore, the results for age and work experience were analyzed separately and are accordingly presented in Table 10 and Table 11.

The results of the moderating effect of age among the demographic characteristics of the study participants are presented in Table 10. Age was analyzed to verify the moderating effect by dividing the participants into those under 30 (*n* = 150) and those over 40 (*n* = 150). This classification was made to ensure statistical validity based on the sample size. The results showed a significant moderating effect of age (*p* = 0.002).

When examining the moderating effect of age, effort expectancy showed a significant positive influence on behavioral intention for respondents under 30, with *β* = 0.309 (*p* < 0.001), but no significant effect for those over 40, where *β* = 0.065 (*p* = 0.713). In contrast, the impact of social influence on behavioral intention was significant for both age groups, with *β* = 0.577 (*p* < 0.01) for individuals under 30, and *β* = 0.738 (*p* < 0.001) for those over 40. Notably, while both groups exhibited positive effects, the influence was stronger in the 40+ age group compared to the under-30 group.

These findings support Hypothesis 7 of this study. It confirms the need for a differentiated approach based on age, which can be an essential consideration for organizations when formulating technology adoption strategies.

In addition, the moderating effect of work experience was analyzed by dividing the work experience of the participants in this study by five years. The moderating effect was analyzed as df = 5, CMIN = 32.172, *p* = 0.000, NFI Delta-1 = 0.003, RFI rho-1 = 0.002, and TLI rho2 = 0.003, as shown in Table 11, with the moderating effect being statistically significant.

When analyzing the moderating effect of work experience, effort expectancy was found to have a significant positive effect on behavioral intention for individuals with less than 5 years of experience (*β* = 0.393, *p* < 0.01), but no significant effect for those with more than 5 years of experience (*β* = 0.071, *p* = 0.928), indicating that only the group with less than 5 years of experience demonstrated a significant positive relationship. Social influence, on the other hand, had a significant positive effect on behavioral intention for both groups: *β* = 0.297 (*p* < 0.05) for less than 5 years and *β* = 0.928 (*p* < 0.001) for more than 5 years, with a stronger effect observed in the latter group.

For the effect of facilitating conditions on behavioral intention, a positive influence was observed in the group with less than 5 years of experience (*β* = 0.339, *p* < 0.01), whereas a negative influence was found in the group with more than 5 years of experience (*β* = −0.192, *p* < 0.05). In terms of the relationship between behavioral intention and usage behavior, both groups exhibited a significant positive effect: *β* = 0.883 (*p* < 0.001) for less than 5 years of experience and *β* = 0.849 (*p* < 0.001) for more than 5 years, with a slightly stronger effect observed in the latter group.

The results of these analyses support Hypothesis 8 of this study. This shows that technology adoption strategies should vary by work experience, which can be an essential consideration for organizations when formulating their technology adoption strategies.

The results of the previous hypothesis testing are shown in Table 12.

## 4. Discussion

The purpose of this study was to analyze the factors that influence the adoption and usage behavior of generative AI systems and suggest effective adoption strategies for them in Korean companies. Based on the UTAUT model, we empirically analyzed the effects of variables such as performance expectancy, effort expectancy, social influence, and facilitating conditions on intention to use and usage behavior.

In analyzing research hypothesis H1, which posits that performance expectancy positively influences the intention to use generative AI systems, we found that performance expectancy did not have a statistically significant effect on the intention to adopt these systems. This indicates that the expectation of enhanced job performance through generative AI is not a decisive factor in driving adoption intentions. While prior research has established performance expectancy as a critical element in technology acceptance [17], with [54] reporting a strong positive effect in small and medium-sized enterprises (SMEs) where immediate performance gains and cost savings from digitalization are highly valued, our findings align with studies by [55,56], which similarly found that performance expectancy had no significant impact. This suggests that in the early stages of generative AI adoption, users may prioritize learning and adaptation over immediate performance outcomes. For Korean firms adopting generative AI systems, this may imply that initial training and adaptation are more critical factors than performance expectancies. Additionally, differences in the respondent sample and the maturity of technology adoption must be considered. While [54]’s study focused on SMEs with high digitalization maturity, our study reflects the early phase of AI technology adoption, where performance expectations may still be underdeveloped.

Research hypothesis H2, that effort expectancy will positively impact behavioral intention, was significant. This means that the more accessible the generative AI system is to use and the more compatible it is with existing systems and processes, the higher the intention to adopt. These results show that the ease of use of the technology is essential and is consistent with existing research [19]. In addition, recent research has also shown that effort expectancy has a significant impact on technology adoption intention [18]. This suggests that ease of use is essential for users when adopting new technology.

Research hypothesis H3, which states that social influence will positively affect behavioral intention, was also significant. This indicates that positive perceptions and support from others, such as coworkers and supervisors, are essential in increasing behavioral intention. This is consistent with existing research and reaffirms the importance of social influence in technology acceptance [57]. A recent study also reported that social influence significantly impacts technology adoption intention [24]. This suggests that positive perceptions and support within the organization play an essential role in the success of technology adoption, and the collectivistic culture of Korean companies, in particular, may reinforce these findings.

Prior studies analyzing the impact of facilitating conditions on behavioral intention have generally reported positive findings. Ref. [36] identified facilitating conditions as a crucial factor in enhancing behavioral intention to use e-wallets, while [18] underscored their central role in technology acceptance models. Similarly, ref. [27] highlighted the necessity of technical support for the successful implementation of AI-based services. However, in this study, research hypothesis H4, which posited that facilitating conditions would significantly influence behavioral intention, did not yield a meaningful effect. This indicates that, in the early stages of technological adoption, ease of use and social support may take precedence over facilitating conditions. Furthermore, perceptions of facilitating conditions could vary depending on organizational size and technological maturity. Interestingly, ref. [58] also found that the influence of facilitating conditions on initial adoption intention was not substantial, aligning with the results of our study.

Research hypothesis H5 was significant, which is that behavioral intention will positively influence actual usage behavior. This means that the higher behavioral intention, the more people will use generative AI systems. This is consistent with existing research, which shows that behavioral intention is more likely to lead to actual behavior [43]. Recent studies have shown similar results. For example, ref. [59] highlighted the critical role of behavioral intention for the success of technology adoption in digital transformation and HRM (Human Resources Management). They found that higher behavioral intention leads to actual behavior. Ref. [60] also found that when AI is integrated with HRM, higher intention to use translates into actual use behavior. Ref. [61] analyzed long-term changes in ChatGPT usage behavior and demonstrated that behavioral intention is essential in reinforcing actual behavior. These studies suggest that increasing behavioral intention has an essential impact on actual technology use, which means that the same can be applied to Korean companies.

Research hypothesis H6, that behavioral intention would significantly mediate the effects of performance expectancy, effort expectancy, social influence, and facilitating conditions on usage behavior, was partially accepted. Social influence significantly mediated behavioral intention, but no mediating effect was found for performance expectancy, effort expectancy, and facilitating conditions. This means that behavioral intention is a significant mediator, but only for some variables. This suggests that behavioral intention mediates some variables but not consistently across all variables. A recent study also reported that behavioral intention only mediates some variables [18], similar to our results.

Research hypothesis H7, which states that there is a moderating effect of age, was significant. Social support has a more substantial effect on technology acceptance the older an individual’s age, while effort expectancy is only significant for those in their 30s and younger, and the influence of behavioral intention on usage behavior is more substantial for those in their 40s and older. This suggests that technology adoption strategies should be differentiated by age, indicating that age may have a differential impact on the adoption and use of generative AI systems. This is consistent with existing research, which suggests that age is a vital variable in technology acceptance [32]. Recent studies have also found that age has a differential impact on technology adoption intention. For example, ref. [36] reports that age significantly affects the adoption of digital payment systems, suggesting the need for age-specific technology adoption strategies. In addition, ref. [62] analyzes the impact of age on intention to adopt mobile banking, emphasizing that age plays a vital role in technology acceptance. Ref. [63] examined the intention to adopt technology among older adults and specifically addressed the impact of age on technology acceptance. These studies suggest that age-specific technology adoption strategies are necessary and should be considered by Korean companies.

Research hypothesis H8, that there is a moderating effect of work experience, was also significant. Employees with more than five years of experience were more sensitive to social support, which increased their intention to adopt technology; meanwhile, facilitating conditions positively affected employees with less than five years of experience and negatively affected employees with more than five years of experience. Effort expectancy was only significant for those with less than five years of experience. The influence of behavioral intention on usage behavior was more substantial for those with more than five years of experience. This suggests that technology adoption strategies should be differentiated by experience and that work experience may have a differential impact on the adoption and use of generative AI systems. In terms of social influence and behavioral intention, more experienced employees may have more positive attitudes toward adopting new technologies [64]. Recent studies have also reported differential effects of work experience on technology adoption intention. For example, ref. [65] reports that organizational maturity plays a significant role in technology adoption intention, indicating that more experienced employees have more positive attitudes. Ref. [66] analyzes intention to adopt electronic payment systems using the UTAUT model and show that employees with more experience are more likely to adopt new technologies. Furthermore, ref. [67] revisits the UTAUT model and presents a revised theoretical model of technology acceptance intention, emphasizing that personal background, such as work experience, significantly impacts technology adoption intention. Ref. [68] found that experienced employees are more receptive to digital technology and remote work, primarily due to their autonomy and strong networking skills. These employees understand their tasks well, execute them efficiently, and effectively utilize their networks, making them better suited for the challenges of working from home.

This suggests that technology adoption strategies should be differentiated by experience and that work experience may have a differential impact on the adoption and use of generative AI systems. Additionally, the adoption of artificial intelligence may either facilitate or hinder employees’ proactive career behavior, depending on the level of organizational support provided, particularly for those with varying levels of experience [37].

## 5. Conclusions

This study deepens the understanding of technology acceptance theory through an empirical investigation into the adoption and usage behavior of generative AI systems, using the UTAUT model as its conceptual framework. The research holds particular significance by systematically analyzing the factors influencing the adoption and usage of these AI systems within the specific context of Korean companies, which are in the early stages of integrating this technology. The findings offer practical insights into technology acceptance, shedding light on the distinct challenges and considerations that firms face during the initial phases of AI technology implementation.

The hypothesis testing results reveal that performance expectancy does not significantly influence behavioral intention, suggesting that it may not be a decisive factor in the early adoption of generative AI. In contrast, effort expectancy and social influence were found to significantly enhance behavioral intention, highlighting the role of ease of use and the impact of social dynamics in shaping adoption behavior. Facilitating conditions, however, did not show a significant effect on behavioral intention. Moreover, behavioral intention had a positive effect on actual usage behavior, partially confirming its mediating role between key adoption factors, including performance expectancy, effort expectancy, social influence, facilitating conditions, and actual use. The moderating effects of age and work experience were also found to be significant, offering additional insight into how these variables interact with technological adoption.

This research distinguishes itself from previous studies by focusing on the early stages of generative AI system adoption, an area at the forefront of technological innovation. Unlike earlier research that predominantly examined established IT systems or specific mature technologies, this study demonstrates that performance expectancy may be less influential in the nascent stages of generative AI adoption. This provides a fresh perspective on how user acceptance intentions evolve alongside the maturity of a given technology. Furthermore, while many prior studies emphasize technical factors, this research brings to the forefront the importance of psychological and organizational influences, such as effort expectancy and social influence, in driving AI adoption behavior. By applying the UTAUT model ([17]) to the current context of generative AI adoption, this study makes a substantial academic contribution to identifying the critical variables that influence AI technology adoption in today’s rapidly advancing technological landscape.

In practical terms, to adopt and apply generative AI systems, Korean companies should first improve users’ ease of use by providing user-centered and intuitive interfaces and running a systematic user training program to ease initial adaptation. In addition, a positive user experience should be fostered through ongoing attention and internal meetings and workshops to garner support from peers and supervisors.

They also need a differentiated and customized approach that considers age and experience. Younger and less experienced employees should be provided with more training and support. In contrast, more experienced employees should emphasize the interest and support of their supervisors and how generative AI systems can integrate with their existing work, for example, by regularly collecting feedback and updating the system to measure performance, sharing success stories to motivate others in the organization, and looking for ways to optimize the use of the system.

Therefore, these research findings provide important implications for companies in formulating technology adoption strategies during the Digital–AI transformation process. This study provides important insights into employee behavior in Digital–AI Transformation by analyzing the factors that influence employees’ technology acceptance intentions during the adoption of generative AI systems. In particular, the finding that effort expectancy and social influence have a significant impact on employees’ technology acceptance behaviors in the early stages of technology adoption provides essential information for understanding how employees will accept new technologies and apply them to their work during the Digital–AI Transformation. This is the first empirical study to systematically analyze the intention to adopt generative AI technology based on [17]’s UTAUT model in Korean companies, and empirically identifies the impact of the latest AI innovations on employees’ behavior. Digital–AI Transformation is only as good as the attitudes and behaviors of employees toward new technologies. This study empirically analyzes the key determinants of employee behavior in the early stages of AI adoption and suggests that companies should strengthen employee training, organizational support systems, and ease of use of technology to successfully drive AI innovation. In particular, the findings that effort expectancy and social influence are the main determinants of employee behavior provide useful insights for formulating strategies to drive employee behavior changes during digital transformation.

Limitations of this study include the limited sample size of the survey, which was primarily Korean corporate employees, which may limit the applicability of the results to organizations in other countries or cultures. In addition, the cross-sectional research design of this study limits the ability to identify changes in variables over time or causal relationships. The study data was collected through self-reported surveys, which may be subject to bias due to respondents’ subjective judgment. This study provides a comprehensive empirical analysis of the adoption and usage behavior of generative AI systems in Korean companies, offering valuable insights into the factors influencing early-stage technology adoption. While the findings largely supported the initial hypotheses, indicating that respondents generally understood the survey questions, there remains the possibility that some technical terms were not fully comprehended by all participants. To enhance the clarity and reliability of future studies, it would be beneficial to implement pre-validation methods, such as cognitive interviews or pilot feedback surveys, to ensure that respondents have a consistent understanding of key terminology. Validating these terms in advance would further improve the design of survey instruments and increase the robustness of the results.

This research also revealed that certain hypotheses, notably those concerning performance expectancy (H1) and facilitating conditions (H4), did not achieve statistical significance. This could reflect either a lack of strong perceptions regarding the potential performance benefits of generative AI systems or insufficient support structures within the organizations studied. Future research should aim to refine the measurement of these constructs by developing more specific and targeted survey items. Additionally, conducting comparative studies across different industries and organizational sizes may provide greater insights into the contextual factors that influence AI adoption. Employing longitudinal research designs would further enrich understanding by capturing how performance expectations and facilitating conditions evolve over time, thereby offering deeper insights into their effects on technology acceptance and usage behavior. Expanding the geographic and industrial scope of such research would contribute to a more nuanced understanding of AI adoption dynamics.

While this study examined both the learning and adoption processes of AI technology in an integrated manner, future research should distinguish between these two stages. Separating the learning phase from post-adoption usage would allow for a clearer analysis of how each process independently affects technology acceptance and usage behavior. This distinction would also enable companies to develop more tailored strategies that support both initial learning and sustained usage, thereby optimizing the adoption process across its lifecycle.

The study applied the Unified Theory of Acceptance and Use of Technology (UTAUT) model to explain the adoption of generative AI systems. Although other theoretical frameworks such as the Technology Acceptance Model (TAM) and Diffusion of Innovation (DOI) are widely used in the field, UTAUT offers a more comprehensive framework for understanding the complex interactions between factors like performance expectations, effort expectancy, social influence, and facilitating conditions. Furthermore, UTAUT’s ability to account for moderating variables such as age and work experience makes it particularly well-suited to studies within organizational contexts. Future research could benefit from applying and comparing alternative models such as TAM or DOI to further expand understanding of the diverse factors that drive technology adoption in various organizational and cultural settings.

## Figures and Tables

**Figure 1 behavsci-14-01035-f001:**
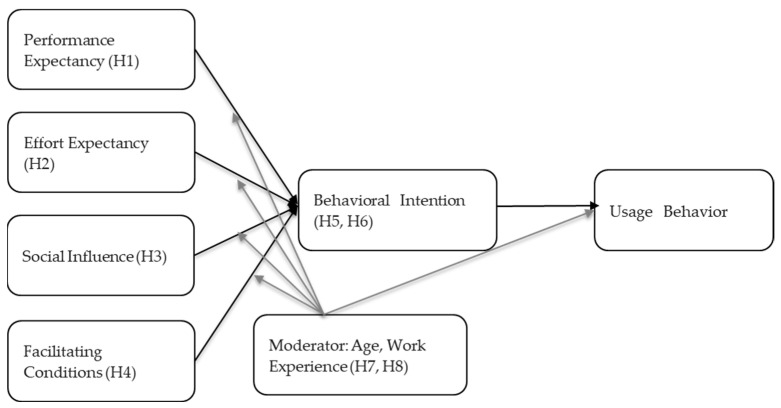
Research Model.

**Table 1 behavsci-14-01035-t001:** Demographics.

Survey Items		Frequency	Percentages
Gender	Male	150	50.0
	Female	150	50.0
Age	20–29 years old	75	25.0
	30–39 years old	75	25.0
	40–49 years old	75	25.0
	50+ years old	75	25.0
Education	High School Graduate or Less	32	10.7
	Community College Graduate	48	16.0
	Graduated from college	186	62.0
	Graduate school or higher	34	11.3
Job Title	CEO & Executives	12	4.0
	Department Head	29	9.7
	Manager, Assistant Director	67	22.3
	Associate, Principal, Assistant	188	62.7
	Others	4	1.3
Type of company (Framework Act on SMEs in South Korea)	Large Domestic Enterprises	32	10.7
	Domestic Midsize (300–1000 employees)	47	15.7
	Domestic Small Business (50–300 employees)	84	28.0
	Domestic small business (50 employees or less)	109	36.3
	Domestic ventures and startups	6	2.0
	Other	22	7.3
Work experience	Less than 5 years	120	40.0
	More than 5 years–less than 10 years	75	25.0
	More than 10 years–Less than 20 years	70	23.3
	More than 20 years	35	11.7
Experience working directly on AI-related tasks	Yes	62	20.7
	No	210	70.0
	Not Sure	28	9.3
Type of Industry	Manufacturing	80	26.7
	Financial	10	3.3
	Retail	19	6.3
	Service industry	139	46.3
	Primary industries (agriculture, fishing, etc.) and construction, etc	20	6.7
	Other	32	10.7
Generative AI usage types	No	136	45.3
	ChatGPT Free	144	48.0
	ChatGPT Paid	15	5.0
	Gemini	5	1.7
The level of your corporate system	Personal PC level	175	58.3
	Basic IT system	58	19.3
	Comprehensive IT system with ERP	44	14.7
	Cloud-based IT system	21	7.0
	Other	2	0.7

**Table 2 behavsci-14-01035-t002:** Operational definition of variables.

Variables		Operational Definitions and Concepts of Variables	Previous Thesis Researcher	Number of Questions
Attributes	Demographics	300 employees in a company using a random sampling method that includes a range of job titles and ages from large and small businesses.	[38]	10
Independent	Performance Expectancy	The degree to which individuals believe that using AI service technology will help improve their quality of life.	[17,19,40,41,42,43]	7
	Effort Expectancy	How easy you believe it will be to learn and use AI services.		7
	Social Influence	The extent to which important people around me recognize that I feel compelled to use AI services.		10
	Facilitating Conditions	The extent to which you believe an organized technical environment exists to support the use of AI services.		7
Mediator	Behavioral Intention	Intent to use generative AI systems.	[17,43]	2
Dependent	Usage Behavior	Try and use new ways of working with generative AI systems.	[17,19]	2

**Table 3 behavsci-14-01035-t003:** Descriptive Statistics.

Variables Name	Mean	Standard Deviation	Skewness	Kurtosis
Performance Expectancy	5.34	0.89	−0.45	0.31
Effort Expectancy	4.84	0.95	−0.15	−0.05
Social Influence	4.30	1.18	−0.68	0.45
Facilitating Conditions	3.81	1.39	−0.19	−0.58
Behavioral Intention	4.90	1.17	−0.52	0.70
Usage Behavior	4.45	1.43	−0.63	0.11

**Table 4 behavsci-14-01035-t004:** Exploratory factor analysis and reliability for independent variables.

Items	Components			
	1	2	3	4
Social Influence 3	0.828			
Social Influence 1	0.818			
Social Influence 2	0.818			
Social Influence 4	0.784			
Social Influence 8	0.749			
Social Influence 10	0.744			
Social Influence 9	0.743			
Social Influence 6	0.733			
Social Influence 7	0.728			
Social Influence 5	0.663			
Facilitating Conditions 4		0.843		
Facilitating Conditions 5		0.840		
Facilitating Conditions 3		0.827		
Facilitating Conditions 7		0.820		
Facilitating Conditions 1		0.798		
Facilitating Conditions 2		0.754		
Facilitating Conditions 6		0.650		
Performance Expectancy 6			0.837	
Performance Expectancy 2			0.828	
Performance Expectancy 7			0.783	
Performance Expectancy 4			0.779	
Performance Expectancy 1			0.763	
Performance Expectancy 3			0.747	
Performance Expectancy 5			0.622	
Effort Expectancy 1				0.824
Effort Expectancy 3				0.809
Effort Expectancy 2				0.805
Effort Expectancy 4				0.620
Effort Expectancy 6				0.584
Effort Expectancy 5				0.533
Effort Expectancy 7				0.497
Eigenvalue	7.227	5.517	5.359	3.803
explained variance (%)	23.312	17.796	17.286	12.266
variance criterion (%)	23.312	41.109	58.395	70.661
Cronbach Alpha	0.956	0.955	0.904	0.891
KMO = 0.939, Bartlett χ^2^ = 8503.318, df = 465, *p* < 0.001				

**Table 5 behavsci-14-01035-t005:** Exploratory factor analysis and reliability for mediators, dependent variables.

Survey Items	Components	
	1	2
Behavioral Intention 1	0.878	
Behavioral Intention 2	0.825	
Usage Behavior 1		0.914
Usage Behavior 2		0.802
Eigenvalue	1.747	1.714
explained variance (%)	43.667	42.843
variance criterion (%)	43.667	86.509
Cronbach Alpha	0.811	0.858
KMO = 0.767, Bartlett χ^2^ = 666.985, df = 465, *p* < 0.001		

**Table 6 behavsci-14-01035-t006:** Confirmatory factor analysis and discriminant validity.

Path			*B*	*β*	S.E.	C.R.	*p*	AVE	CCR
Social Influence 5	←	Social Influence	1.000	0.780	Fixed			0.515	0.914
Social Influence 7	←		1.197	0.836	0.074	16.246	***		
Social Influence 6	←		1.230	0.872	0.072	17.184	***		
Social Influence 10	←		1.256	0.822	0.079	15.896	***		
Social Influence 9	←		1.191	0.876	0.069	17.288	***		
Social Influence 8	←		1.111	0.807	0.072	15.514	***		
Social Influence 4	←		1.209	0.871	0.070	17.157	***		
Social Influence 1	←		1.160	0.768	0.080	14.556	***		
Social Influence 2	←		1.123	0.779	0.076	14.816	***		
Social Influence 3	←		1.178	0.791	0.078	15.129	***		
Performance Expectancy 3	←	Performance Expectancy	1.000	0.777	Fixed			0.606	0.902
Performance Expectancy 1	←		0.949	0.841	0.060	15.839	***		
Performance Expectancy 4	←		0.912	0.712	0.071	12.926	***		
Performance Expectancy 7	←		0.959	0.769	0.068	14.179	***		
Performance Expectancy 2	←		1.074	0.882	0.064	16.809	***		
Performance Expectancy 6	←		1.015	0.802	0.068	14.934	***		
Facilitating Conditions 6	←	Facilitating Conditions	1.000	0.735	Fixed			0.563	0.900
Facilitating Conditions 2	←		1.219	0.859	0.079	15.403	***		
Facilitating Conditions 1	←		1.357	0.905	0.083	16.337	***		
Facilitating Conditions 3	←		1.320	0.901	0.081	16.245	***		
Facilitating Conditions 5	←		1.407	0.901	0.087	16.244	***		
Facilitating Conditions 4	←		1.330	0.903	0.082	16.294	***		
Facilitating Conditions 7	←		1.292	0.866	0.083	15.540	***		
Effort Expectancy 5	←	Effort Expectancy	1.000	0.848	Fixed			0.546	0.826
Effort Expectancy 6	←		0.888	0.783	0.058	15.329	***		
Effort Expectancy 4	←		0.967	0.830	0.059	16.523	***		
Effort Expectancy 1	←		0.786	0.644	0.066	11.847	***		
Behavioral Intention 2	←	Behavioral Intention	1.000	0.864	Fixed			0.585	0.737
Behavioral Intention 1	←		0.808	0.796	0.051	15.825	***		
Usage Behavior 2	←	Usage Behavior	1.000	0.947	Fixed			0.572	0.726
Usage Behavior 1	←		0.900	0.796	0.052	17.351	***		
*χ*^2^(df = 416, *n* = 300) = 1021.80, *p* = 0.000, CMIN/df = 2.456. CFI = 0.929, RMR = 0.099, TLI (Tucker-Lweis) = 0.921, IFI = 0.929, RMSEA = 0.070									

*** *p* < 0.001.

**Table 7 behavsci-14-01035-t007:** Correlation analysis.

Variables Name	Performance Expectancy	Effort Expectancy	Social Influence	Facilitating Conditions	Behavioral Intention	Usage Behavior
Performance Expectancy	1					
Effort Expectancy	0.614 **	1				
Social Influence	0.392 **	0.419 **	1			
Facilitating Conditions	0.193 **	0.318 **	0.725 **	1		
Behavioral Intention	0.466 **	0.500 **	0.653 **	0.462 **	1	
Usage Behavior	0.295 **	0.375 **	0.659 **	0.608 **	0.704 **	1

** *p* < 0.01.

**Table 8 behavsci-14-01035-t008:** Hypothesis testing results.

Number	Hypothesis			*B*	*β*	S.E.	C.R.	*p*
H1	Performance Expectancy	→	Behavioral Intention	0.098	0.072	0.094	1.044	0.297
H2	Effort Expectancy	→	Behavioral Intention	0.211	0.174	0.085	2.494	0.013 *
H3	Social Influence	→	Behavioral Intention	0.793	0.662	0.102	7.771	***
H4	Facilitating Conditions	→	Behavioral Intention	0.040	0.037	0.077	0.516	0.606
H5	Behavioral Intention	→	Usage Behavior	1.035	0.863	0.062	16.705	***
*χ*^2^(df = 420, *n* = 300) = 1066.84, *p* = 0.000, CMIN/df = 2.540. CFI = 0.924, RMR = 0.112, TLI (Tucker-Lweis) = 0.916, IFI = 0.925, RMSEA = 0.072								

*** *p* < 0.001, * *p* < 0.05.

**Table 9 behavsci-14-01035-t009:** Results of the mediating effect of behavioral intention.

Independent	Mediator	Dependent	Estimate	95% Confidence Interval		*p*
				LLCI	ULCI	
Performance Expectancy	Behavioral Intention	Usage Behavior	0.062	−0.089	0.171	0.339
Effort Expectancy			0.150	−0.011	0.277	0.063
Social Influence			0.571	0.355	0.806	0.004 **
Facilitating Conditions			0.032	−0.145	0.218	0.717

** *p* < 0.01.

**Table 10 behavsci-14-01035-t010:** Results of verifying the effect of moderation by age.

Hypothesis			30s and Under (*n* = 150)				40+ (*n* = 150)			
			*B*	S.E.	*β*	C.R.	*B*	S.E.	*β*	C.R.
Performance Expectancy	→	Behavioral Intention	0.013	0.106	0.012	0.120	0.230	0.151	0.142	1.523
Effort Expectancy	→	Behavioral Intention	0.319	0.111	0.309	2.886 **	0.088	0.123	0.065	0.713
Social Influence	→	Behavioral Intention	0.491	0.109	0.577	4.508 **	1.177	0.185	0.738	6.362 ***
Facilitating Conditions	→	Behavioral Intention	0.085	0.102	0.095	0.841	−0.063	0.114	−0.049	−0.551
Behavioral Intention	→	Usage Behavior	1.176	0.122	0.869	9.660 ***	0.914	0.069	0.839	13.171 ***
df = 5, CMIN = 19.133, *p* = 0.002, NFI Delta-1 = 0.002, RFI rho-1 = 0.001, TLI rho2 = 0.001										

*** *p* < 0.001, ** *p* < 0.01.

**Table 11 behavsci-14-01035-t011:** Results of verifying the effect of moderation by work experience.

Hypothesis			Less than 5 Years (*n* = 120)				5+ Years (*n* = 180)			
			*B*	S.E.	*β*	C.R.	*B*	S.E.	*β*	C.R.
Performance Expectancy	→	Behavioral Intention	0.010	0.157	0.009	0.063	0.131	0.116	0.085	1.130
Effort Expectancy	→	Behavioral Intention	0.434	0.160	0.393	2.715 **	0.090	0.097	0.071	0.928
Social Influence	→	Behavioral Intention	0.271	0.111	0.297	2.452 *	1.307	0.167	0.928	7.834 ***
Facilitating Conditions	→	Behavioral Intention	0.339	0.113	0.339	3.001 **	−0.219	0.103	−0.192	−2.129 *
Behavioral Intention	→	Usage Behavior	1.103	0.115	0.883	9.607 ***	0.999	0.073	0.849	13.724 ***
df = 5, CMIN = 32.172, *p* = 0.000, NFI Delta-1 = 0.003, RFI rho-1 = 0.002, TLI rho2 = 0.003										

*** *p* < 0.001, ** *p* < 0.01, * *p* < 0.05.

**Table 12 behavsci-14-01035-t012:** Summary of hypothesis testing results.

Number	Hypothesis	
H1	Performance expectancy will have a positive (+) effect on intent to use.	Rejection
H2	Effort Expectancy will have a positive (+) effect on Intent to Use.	Adoption
H3	Social influence will have a positive (+) impact on Behavioral Intentions.	Adoption
H4	Facilitating conditions will have a positive (+) effect on the intent to use.	Rejection
H5	Intent to use will have a positive (+) impact on usage behavior.	Adoption
H6	There will be a significant mediating effect of intention to use on the effects of performance expectancy, effort expectancy, social influence, and facilitating conditions on usage behavior.	Partial Adoption
H7	There will be a moderating effect of age.	Adoption
H8	There will be a moderating effect of work experience.	Adoption

## Data Availability

Data are contained within the article.

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
