# Peer review of "Determinants of Generative AI System Adoption and Usage Behavior in Korean Companies: Applying the UTAUT Model"

_behavsci, 2024, doi:10.3390/bs14111035_

Round 1
Reviewer 1 Report
Comments and Suggestions for Authors
The article presents a very important subject for companies and their issues related to the role of generative AI in the workplace. The literature covers the major developments on the subject, the theoretical framework adopted by the study is appropriate. However, several elements pose serious problems in terms of methodology and data analysis, of which here is one:
1-Adds an analysis of statistical power, especially since the analyzes were conducted using a structural equation model; this requires a larger sample size ;
2-As the study does not aim to develop a measurement scale and is conducted on a single sample, it is more appropriate to conduct confirmatory factor analyzes (CFA) and not an exploratory factor analysis. Thus, in the analyses, authors have to compare null model with other models (model combine performance, effort, social influence, facilitation condition, …) ;
3-For mediation analysis, it is important to break down the total effect into direct effect and indirect effect ;
4. Correct the title of Figure 1 by replacing ‘mediator: Age, Work Experience’ with ‘Moderator: Age, Work Experience for the age variable.
Author Response
We sincerely appreciate your insightful comments to develop our manuscript. We anticipate that our revised manuscript and following responses is appropriate with your comments and suggestions. All changes are written in RED color in the revised manuscript. Please check the attached responses.

Reviewer 2 Report
Comments and Suggestions for Authors
Use the correct format for tables and figures.For example: "<Table 1> Demographic" should be labeled as "Table 1. Demographic".
It is not correct to cite a reference in the format [19,17]. APA references should be listed in ascending numerical order.
Consider using UTAUT2 or justify the use of UTAUT instead of UTAUT2 or TAM.
The "Materials and Methods" chapter should be properly structured, ensuring that the methodology used is presented clearly and in an orderly manner, explaining the procedures step by step. Additionally, it is recommended to use subheadings to organize the information in a more accessible and understandable way.
For the "Materials and Methods" chapter, consider a more descriptive title, like "Methodology".
Author Response

(The authors gave the same response as above.)

Reviewer 3 Report
Comments and Suggestions for Authors
Please refer to the attached file. Thank you.

The language of this manuscript needs to be improved.
Author Response

(The authors gave the same response as above.)

Reviewer 4 Report
Comments and Suggestions for Authors
see attached

Author Response

(The authors gave the same response as above.)

Reviewer 5 Report
Comments and Suggestions for Authors
The introduction of this study lacks clarity regarding the research objectives. It is vital to strengthen this section to clearly convey the necessity of the study and the context of the research problem logically.
Please conduct a more rigorous check of the reliability and validity of the measurement tool used. It is advised to perform factor analysis that includes all variables listed in Tables 4 and 5. Additionally, please investigate for any common method bias. While this currently is not assessing discriminant validity, it should still be verified.
I believe that the inconsistency between the research hypothesis and the analytical model represents the primary weakness of this study. Based on the hypothesis illustrated in Figure 1, Table 10 should also display the effects of Age and Work Experience on Usage Behavior.
To enhance the readability of this paper, please ensure that table lines and content are aligned correctly. There are inconsistencies in statistical terminology (e.g., df, DF, or Df), particularly when directly presenting results from statistical software. Please rewrite it according to the specified format and guidelines. Fonts and sizes must also comply with this journal, Behavioral Sciences' rules.
I would like the authors to exercise greater precision in testing the moderation effects. It is unreasonable to categorize the overall hypothesis testing results simply as acceptance or rejection. I recommend examining the differences in beta coefficients between the two groups.
Please ensure that the references are formatted in accordance with this journal's guidelines. The reference list should include full titles, as recommended by the ACS style guide.
For the preceding thesis, ADOPTION AND USE OF ELECTRONIC INSTRUCTIONAL MEDIA AMONG ACADEMICS IN SELECTED UNIVERSITIES IN SOUTH WEST NIGERIA written by Alabi, A. O. (2016), please double-check that there are no copywrite issues.
Comments on the Quality of English LanguagePlease break up long sentences and keep your terminology consistent.
Author Response

(The authors gave the same response as above.)

Round 2
Reviewer 1 Report
Comments and Suggestions for Authors
Hi,
In their new version, authors did not adress our recommendation particularly those below :
1. As the study does not aim to develop a measurement scale and is conducted on a single sample, it is more appropriate to conduct confirmatory factor analyzes (CFA) and not an exploratory factor analysis. Thus, in the analyses, authors have to compare null model with other models (model combine performance, effort, social influence, facilitation condition, …) ;
2. For mediation analysis, it is important to break down the total effect into direct effect and indirect effect ;
Best regard;
Author Response
Comment1: As the study does not aim to develop a measurement scale and is conducted on a single sample, it is more appropriate to conduct confirmatory factor analyzes (CFA) and not an exploratory factor analysis. Thus, in the analyses, authors have to compare null model with other models (model combine performance, effort, social influence, facilitation condition, …) ;
Response1: Thank you for your valuable comments. We have revised the Exploratory Factor Analysis and Confirmatory Factor Analysis sections of the methodology to include the following additional content.
(Page 8, Line 251)
"
To accomplish the analytical objectives of this study, SPSS 22 and AMOS 21 were employed. Descriptive statistics and bivariate correlation analyses were first conducted in SPSS 22 to explore the basic interrelationships among variables. Reliability analysis was conducted, adopting a Cronbach's Alpha value of 0.6 or higher, as suggested by [45]. Although exploratory factor analysis (EFA) was initially used to preliminarily align factor structures with the theoretical framework, the primary analytical focus was on confirmatory factor analysis (CFA) carried out in AMOS 21. Upon confirming multivariate normality, CFA was executed via the Maximum Likelihood method to assess construct validity, with path coefficients examined to evaluate model fit. For model fit assessment, the hypothesized model, which incorporates performance expectancy, effort expectancy, social influence, and facilitating conditions, was compared against a null model with no assumed relationships among latent variables, ensuring that the hypothesized model better represented the data and corresponded to the study’s theoretical constructs.
To verify the model’s validity further, convergent and discriminant validity were evaluated through Average Variance Extracted (AVE) and Composite Construct Reliability (CCR). The indirect effects of usage intention were analyzed through bootstrapping (500 resamples, with ? < .05), and the moderating effects of age and work experience were assessed using Multi-Group Analysis (MGA). All statistical analyses followed a significance level of ? < .05.
Additionally, detailed fit indices were reported for both the null model and the hypothesized model. Fit indices such as Chi-square (χ²), RMSEA, CFI, TLI, and RMR were analyzed to clearly demonstrate the improvements in the hypothesized model. These results underscore the alignment of the hypothesized model with theoretical expectations, validating that the model integrating performance expectancy, effort expectancy, social influence, and facilitating conditions offers a more accurate and meaningful fit than the null model."
(Page 11, line 350)
"
The convergent validity of the factors selected in this study was demonstrated through exploratory factor analysis and reliability analysis. However confirmatory factor analysis and discriminant validity analysis were conducted to confirm the single dimension of each factor for the measurement items, and the results are shown in Table 6. Compared to the indicators of Model Fit, ?²(??=416, ?=300)=1021.80, ?=.000, CMIN/df=2.456. ???=.929, RMR=.099, ???(Tucker-Lweis)=.921, IFI=. 929, and ?????=.070, confirming that the structural model fit for the research model set in this study meets the criteria. In addition, the average variance extracted (AVE) of this study was selected to be above 5 [51] and the composite construct reliability (CCR) was selected to be above 7 [52, 53], and the results of the analysis showed that they all met the criteria, confirming the discriminant validity of the measurement tools.
In addition, as a result of confirmatory factor analysis, the SMC values of performance expectancy No. 3, effort expectancy No. 2, 3, and 7 were found to be below the criterion value and were deleted. The size of the critical ratio (C.R.) of the structural model estimation after deletion was interpreted as an absolute value of 1.96 or more, as shown in Table 6. As shown in Table 6, the C.R. values of the measured variables were found to be above the criterion of 1.96 and significant at ?<.001, so it is judged that the convergent validity of this research model is proven."
Comment 2: For mediation analysis, it is important to break down the total effect into direct effect and indirect effect ;
Response 2: Thank you for the valuable feedback. We made the following additions to the mediating effects analysis.
(Page 15, line 402)
"
The results of the analysis using bootstrapping to verify the mediating effect of behavioral intention, a parameter of this study, are shown in Table 9. The results indicate that social influence has a significant indirect effect on usage behavior through behavioral intention, with a 95% confidence interval between 0.355 and 0.806 (?<.01). This finding demonstrates that social influence significantly contributes to usage behavior when mediated by behavioral intention, supporting Hypothesis 6 as partially accepted. However, no significant indirect effects were observed for performance expectancy, effort expectancy, and facilitating conditions.
To further clarify the effect pathways, Hypothesis 5, which posits that behavioral intention has a direct effect on usage behavior, was confirmed with a strong and statistically significant outcome (?=0.863, ?<.001). This suggests that behavioral intention has a substantial direct impact on usage behavior, independently of any mediating variables. Thus, as employees' behavioral intention to use generative AI system strengthens, their actual usage behavior proportionally increases.
These findings highlight the importance of both direct and indirect pathways, where the direct effect of behavioral intention on usage behavior plays a primary role, while the indirect pathway through social influence emphasizes the critical role of external social factors in the early stages of generative AI system adoption."
Reviewer 3 Report
Comments and Suggestions for Authors
After revision, the manuscript has been substantively improved.
Author Response
After revision, the manuscript has been substantively improved.:
Response: Thank you for your thoughtful feedback and careful review. We are pleased to hear that the revisions have improved the manuscript. Your insights were invaluable in strengthening the quality of our work.
Reviewer 5 Report
Comments and Suggestions for Authors
The review point was addressed well. Great job on revising the paper. Please double-check for minor editorial errors and ensure consistency, such as table formats and text fonts.
Comments on the Quality of English LanguageThe paper is much improved with academic English expressions, but there are still some unnatural expressions, so I hope that you can improve the overall writing.
Author Response
Comment: The review point was addressed well. Great job on revising the paper. Please double-check for minor editorial errors and ensure consistency, such as table formats and text fonts.
Response: Thank you for your positive feedback and kind words. We appreciate your guidance throughout the revision process. We will carefully double-check for any minor editorial errors and ensure consistency in table formats and text fonts to further enhance the manuscript's quality.